# Integrated hydrometeorological – snow – frozen ground observations in the alpine region of the Heihe River Basin, China

Tao Che[1,2,3], Xin Li[2,3], Shaomin Liu[4], Hongyi Li[1], Ziwei Xu[4], Junlei Tan[1], Yang Zhang[1], Zhiguo Ren[1], Lin Xiao[1], Jie Deng[1,6], Rui Jin[1], Mingguo Ma[5], Jian Wang[1], Xiaofan Yang[4]

[1] Heihe Remote Sensing Experimental Research Station, Key Laboratory of Remote Sensing of Gansu Province, Cold and Arid Regions Environmental and Engineering Research Institute, Chinese Academy of Sciences, Lanzhou 730000, China
[2] Center for Excellence in Tibetan Plateau Earth Sciences, Chinese Academy of Sciences, Beijing 100101, China
[3] Institute of Tibetan Plateau Research, Chinese Academy of Sciences, Beijing 100101, China
[4] State Key Laboratory of Earth Surface Processes and Resource Ecology, Faculty of Geographical Science, Beijing Normal University, Beijing 100875, China
[5] Chongqing Engineering Research Center for Remote Sensing Big Data Application, School of Geographical Sciences, Southwest University, Chongqing 400715, China
[6] Jiangsu Center for Collaborative Innovation in Geographical Information Resource Development and Application, Nanjing 21003, China

*Correspondence to*: Xin Li (xinli@itpcas.ac.cn)

**Abstract.** The alpine region is important in riverine and watershed ecosystems as a contributor of freshwater, providing and stimulating specific habitats for biodiversity. In parallel, recent climate change, human activities and other perturbations may disturb hydrological processes and eco-functions, creating the need for next-generation observational and modeling approaches to advance a predictive understanding of such processes in the alpine region. However, several formidable challenges, including the cold and harsh climate, high altitude and complex topography, inhibit complete and consistent data collection where/when needed, which hinders the development of remote sensing technologies and alpine hydrological models. The current study presents a suite of datasets consisting of long-term hydrometeorological, snow cover and frozen ground data for investigating watershed science and functions from an integrated, distributed and multiscale observation network in the upper reaches of the Heihe River Basin (HRB) in China. Meteorological and hydrological data were monitored from an observation network connecting a group of automatic meteorological stations (AMSs). In addition, to capture snow accumulation and ablation processes, snow cover properties were collected from a snow observation superstation using state-of-the-art techniques and instruments. High-resolution soil physics datasets were also obtained to capture the freeze-thaw processes from a frozen ground observation superstation. The updated datasets were released to scientists with multidisciplinary backgrounds (*i.e.*, cryospheric science, hydrology, and meteorology), and they are expected to serve as a testing platform to provide accurate forcing data and validate and evaluate remote sensing products and hydrological models for a broader community. The datasets are available from the Cold and Arid Regions Science Data Center at Lanzhou https://doi.org/10.3972/hiwater.001.2019.db.

## 1. Introduction

Water resources in the alpine region are headwaters to sustain downstream ecosystems. However, perturbations induced by nature/climate change and human activities in recent years have significantly altered hydrological processes and eco-functions (Li *et al.*, 2018b). Accurate estimation and prediction of hydrological processes and their key impacts has since become crucial (Pomeroy *et al.*, 2007; Chen *et al.*, 2014; Li *et al.*, 2018c). Process-based alpine hydrological models (*e.g.*, the Geomorphology-Based Eco-Hydrological Model (GBEHM), Yang *et al.*, 2015; the Water and Energy Budget-based Distributed Hydrological Model (WEB-DHM), Wang *et al.*, 2010; the Cold Regions Hydrological Model (CRHM), Pomeroy *et al.*, 2007; the Cryospheric Basin Hydrological Model (CBHM), Chen *et al.*, 2018) are feasible to advance a fundamental understanding of the hydrological cycle and its individual components, *i.e.*, separating the contributions from processes such as snow melting, freeze-thaw, precipitation, evapotranspiration, runoff, and determining their spatiotemporal distributions across scales. Unfortunately, the scarcity of observation data in the alpine region, due to the difficulties of access and technological barriers,

has hindered alpine hydrological modeling and associated research yet motivated the development of next-generation ecosystem observation networks and experiments (Hrachowitz *et al*., 2013). In comparison with observations, using remote sensing data combined with data assimilation could improve the prediction of hydrological processes (Schmugge *et al*., 2002; Clark *et al*., 2007). However, due to the complexities of the earth system, there exist various sources of uncertainties in remote sensing data (especially in the alpine region), which have to be validated and calibrated (Hall *et al*., 2007; Jackson *et al*., 2010; Che *et al.,* 2012; Frei *et al*., 2012; Dai *et al.,* 2017). In summary, to fill the knowledge gap and promote alpine region hydrology research, an integrated, distributed and multiscale observation dataset is essential and expected to provide accurate forcing data for hydrological modeling, validate remote sensing data, allow the evaluation of distributed models and ultimately improve a predictive understanding of alpine hydrological processes and ecosystem functions.

In alpine hydrology, in addition to consistent hydrometeorological data obtained from distributed meteorological stations, snow cover and frozen ground are two important indexes and driving forces that influence hydrological processes (Cline, 1997; Dewalle and Rango, 2008; Walvoord and Kurylyk, 2016). The maximum snow water equivalent (SWE) before ablation determines the storage of snowmelt – a major source of freshwater in the alpine region (Chahine, 1992), while freeze-thaw cycles (FTCs) and soil moisture within the active layer alter water infiltration and, consequently, surface runoff and groundwater (Shanley and Chalmers, 1999; Hardy *et al*., 2001). Several representative snow observation stations were established throughout the world to collect snowpack data during seasonal changes at the catchment/watershed scale (Dewalle and Rango, 2008; Kinar and Pomeroy, 2015) in the USA (the Reynolds Creek experimental watershed, Nayak *et al*., 2010; the Sleepers River Basin, Pellerin *et al*., 2012; the Hubbard Brook Basin, Hardy *et al*., 2001; the Loch Vale, Balk and Elder, 2000; the Green Lakes Valley, Caine, 1995), Canada (the Marmot Creek Research Basin, DeBeer and Pomeroy, 2009), and Europe (the Swiss Alps, Dovas, Beniston *et al*., 2003; the Col de Porte experimental site, Morin *et al*., 2012). To promote alpine hydrology research, the International Network for Alpine Research Catchment Hydrology (INARCH) was launched in 2015 by the Global Energy and Water Exchanges (GEWEX) project of the World Climate Research Programme (WCRP), involving 18 catchments around the world (Pomeroy *et al*., 2015). The INARCH has since connected individual observatories into an international network and data-share platform to lead frontier research on alpine region hydrometeorology and snow observation. Another community-based observation network, the Circumpolar Active Layer Monitoring (CALM) network, was initiated in the early 1990s to observe the response of the active layer and near-surface permafrost to climate change (Brown *et al*., 2000). The sites of the CALM network are located not only in the Arctic and Antarctic regions but also in several mid-latitude mountainous regions. The observation infrastructure is designed to include standard active layer and near-surface permafrost measurements, with snow cover, soil moisture, and ground subsidence measured simultaneously at selected sites (Brown *et al*., 2000). In addition, the Global Terrestrial Network for Permafrost (GTN-P) is the primary international program targeted at monitoring permafrost parameters. GTN-P was developed in the 1990s by the International Permafrost Association (IPA) under the Global Climate Observing System (GCOS) and the Global Terrestrial Observing System (GTOS), with the long-term goal of obtaining a comprehensive understanding of the spatial structure, trends and variability of changes in the active layer thickness and permafrost temperature (Streletskiy *et al*., 2017). The CALM and GTN-P have shared stations in their networks and are considered two representative initiatives focusing on frozen ground observation.

The Heihe River Basin (HRB) is the second largest inland river basin in China and is known for its heterogeneous landscapes, diverse ecosystems, unique geographical characteristics and climate change over recent decades. The HRB is mainly composed of glaciers, snow cover, frozen ground, alpine meadows, forests, irrigated crops, riparian ecosystems and deserts (distributed along an altitudinal gradient), which makes it an ideal field site for hydrometeorological research and has motivated and initiated the establishment of China's first basin-scale integrated observatory network (Li *et al*., 2009; Li *et al*., 2013; Li *et al*., 2017; Liu *et al*., 2018). As a major component of the ecohydrological processes in the arid/semi-arid HRB, alpine hydrological processes, especially those related to snow cover and frozen ground in the upper reaches, have marked impacts on runoff in the mountainous regions, which then regulate agricultural development in the middle reaches and the ecosystems of the lower

reaches (Li *et al*., 2018c). To characterize the dynamic alpine hydrological processes influenced by natural variability and recent human activity, extensive research has been conducted in the upper HRB, including both observation and modeling efforts. Nevertheless, long-term observations in the upper reaches of the HRB were often conducted either in a limited range (*e.g.*, small catchment, Chen *et al*., 2014) or focused on single elements (*e.g.*, frozen ground, Peng *et al*., 2016; forest hydrology, He *et al*., 2012 and Wang *et al*., 2013). Several distributed hydrological models have been utilized to predict altered hydrologic processes under various climate scenarios (Wang *et al*., 2010; Zhang *et al*., 2017; Gao *et al*., 2018). However, the above modeling efforts suffered from 1) the availability of the forcing data (only two weather stations operated by the Chinese Meteorological Administration functioned in the upper reaches of the HRB) and 2) a lack of high-quality snow and frozen ground data for parameterization. To overcome the above issues, a comprehensive observation network for alpine hydrology was built in the upper reaches of the HRB since 2013 during the Heihe Watershed Allied Telemetry Experimental Research (HiWATER, Li *et al*., 2013). Composed of seven standard automatic meteorological stations (AMSs), one snow superstation and one frozen ground superstation, the observation network serves as an integrated research platform aiming to provide prominent datasets (*e.g.*, hydrometeorology, snow, and frozen ground) of the hydrometeorological processes in the upper reaches of the HRB, which is expected to support alpine region hydrological model development and simulations along with remote sensing observation. Since 2015, the HRB alpine observation network has joined the INARCH (www.usask.ca/inarch/), which also built a solid foundation for international collaborations.

This paper introduces the infrastructure of the integrated alpine hydrology observation network in the HRB and the complete datasets collected in recent years. The experimental site and design are summarized in Section 2. A brief introduction of the datasets including data availability and access is provided, followed by detailed descriptions of hydrometeorological, snow and frozen ground observations given in Section 3 with subsequent discussions and data analysis. Conclusions with future perspectives are summarized in Section 4.

## 2. An integrated hydrometeorological – snow – frozen ground observation network in the upper reaches of the HRB

### 2.1 Site descriptions

The integrated hydrometeorological – snow – frozen ground observation network was established in an alpine region in the semi-arid region of northwestern China, with a size of 10,009 km$^2$ ranging from 1674 m to 5108 m in altitude (37.72°-39.09°N, 98.57°-101.16°E, Figure 1). It is located in the upper reaches of the HRB (143,200 km$^2$) and the Qilian Mountains (at the intersection of the Qinghai-Tibet Plateau, QTP, and the Mongolian Plateau, MP). The unique geographical characteristics of the HRB (high altitude, complex terrain, various ecosystems and harsh climate) with the widespread snow cover and frozen ground make it an ideal field site for alpine research yet pose great challenges to observation. Detailed descriptions of the study site are provided as follows.

The study site exhibits complex and dynamic hydrometeorological characteristics. The western region is dominated by the mid-latitude westerlies, while the eastern region is influenced by the southwest- and southeast-Asian monsoons. Due to its regional atmospheric circulation and topography, the annual precipitation in the area decreases from the east to the west, with an average of 510 mm (454.7 mm rainfall and 65.3 mm snow) (Li *et al*., 2018c). From low to high altitudes, the land surface of the study area is highly heterogeneous, including alpine grasslands (dominant), alpine shrubs, alpine meadows, tundra, deserts and forest steppes. The soil texture is mainly composed of loam, with silt loam near the surface and sandy soil in deeper layers. In the area covered by grasses and forests, the organic content is very high within the top 30 cm of the soil layer, which impacts the energy and water exchange between the land and atmosphere (Chen *et al*., 2012). The land surface is frozen during the winter across the entire experimental area. The lower limit of the permafrost is between approximately 3700 m and 3800 m, while the rest is seasonally frozen ground. In recent years, with climate warming, permafrost degradation has significantly affected runoff and the carbon cycle (Peng *et al*., 2016; Gao *et al*., 2018). Snow cover is widely distributed with unique

characteristics in the study area. In the high-altitude mountainous regions (elevation > 3800 m), influenced by the local microclimate and low temperature, snowfall could occur in any season. Temporary snow is the major snow cover type in the region at middle and low elevations because strong solar radiation and high air temperature lead to rapid melting and sublimation of the snow. In January and February, snowfall events were rare in historical records due to the relatively low moisture in the atmosphere, while spring and autumn (*i.e.*, March to May, October to November) are considered two main snowfall seasons. Drifting snow is also commonly observed in the region, which may lead to the redistribution of snowpack in high elevation regions (Essery *et al.*, 1999; Li *et al.*, 2014).

## 2.2 Observation infrastructure

Considering the characteristics of the study site, an integrated, distributed and multiscale hydrometeorological – snow – frozen ground observation network has been established with seven AMSs and two superstations (Table 1), with specific scientific focus on the hydrometeorological processes at the basin (the current study site) and sub-basin (the Babao River Basin in Figure 1) scales, as well as two key impact factors in the alpine hydrological process: snow cover and frozen ground. At the basin scale, the observation aims to collect data for investigating the meteorological driving forces and for validation of the alpine hydrological models. At the sub-basin scale, small-scale observations and measurements focus on data such as precipitation, soil temperature and moisture, which are used to develop and improve hydrologic models, as well as to validate remote sensing products at medium to coarse resolutions (Jin *et al.*, 2014). At the two superstations for snow cover and frozen ground observations, remote sensing products and land surface model can be further validated using fine-resolution data.

For hydrometeorological observation, topographic characteristics (elevation, terrain and landscape) were fully considered in the location of the stations/sites. In total, seven AMSs, as well as the frozen ground and snow superstations (also incorporating meteorological observations), were established in the study area during the intensive observation period (IOP: 2013-2014) to obtain the spatially distributed meteorological variables. After the IOP, four typical AMSs were selected and retained for continuous observation, which is expected to provide long-term datasets. The observed meteorological variables include wind (speed and direction), air temperature, humidity, infrared temperature, air pressure, four-component radiation, and precipitation. It is noted that all the sensors were installed in the same way at the same height at each station to guarantee consistency. A typical layout of the AMSs installed at the Dashalong station is shown in Figure 2a.

Snow cover is a prominent feature in the study site. Accumulation and depletion processes were measured automatically at the Yakou snow superstation (Figure 2b). The observed components included snow depth, snow water equivalent (SWE, measured by GammaMONitor, GMON), albedo of the snow surface and blowing (drifting) snow flux (measured by FlowCapt). Solid precipitation was recorded based on a weighing bucket precipitation gauge with a double fence intercomparison reference (DFIR) recommended by the World Meteorological Organization (WMO). In addition, meteorological variables, soil temperature and moisture were also observed.

Soil temperature and moisture were measured within six layers at each AMS, while evapotranspiration (ET) was observed by eddy covariance (EC) at three AMSs located in permafrost, seasonally frozen ground and a transient zone to observe freeze-thaw (FT) processes. At the sub-basin scale, a wireless sensor network (WSN) with 40 nodes was established to capture seasonal changes in soil temperature and moisture (more details and data can be found in Jin *et al.*, 2014). At the A'rou frozen ground superstation (Figure 2c), soil temperature and moisture profiles were intensively measured in eighteen layers to a depth of 3.2 m (in a nested pattern with more layers in the topsoil) to obtain the soil hydrothermal features under freeze-thaw cycles (FTCs). In addition, thermal conductivity and hydraulic conductivity in the topsoil were also measured to observe the dynamic hydrothermal processes within.

Table 2 summarizes all the variables, sensors and measuring locations at the seven AMSs, Yakou snow superstation, and A'rou frozen ground superstation.

## 3. Data descriptions

### 3.1 Dataset availability and access

All datasets presented in this paper have been released and available for free download from the Cold and Arid Regions Science Data Center at Lanzhou (https://doi.org/10.3972/hiwater.001.2019.db). A specific directory was designated for each observation station with data classified into 3 categories: hydrometeorological data, snow cover data and frozen ground data. Short descriptions were also provided for each dataset. Auxiliary data include site descriptions, *e.g.*, a watershed digital elevation model (DEM), shapefiles of the watershed boundary and the station locations.

### 3.2 Meteorological data

Distributed meteorological data were obtained from seven AMSs, most of which were built on flat ground, with two stations on the north-facing and south-facing slopes. All AMSs recorded precipitation, direction and speed of wind, air temperature and humidity, surface air pressure, upward and downward radiation of both short and long waves (four components) and land surface temperature (LST). All sensors (listed in Table 2 with manufacturers, models, and specifications) were calibrated and intercompared before being mounted. The sampling frequencies, reference heights and directions of these sensors at all stations were identical to keep the consistency of the data. For more detailed observations, wind speed, air temperature and humidity were recorded at 1, 2, 5, 10, 15 and 25 m at the A'rou frozen ground superstation. Three eddy covariance (EC) systems were installed to measure evapotranspiration (ET) at the Yakou snow superstation (built on largely distributed permafrost), the A'rou frozen ground superstation (built on seasonal frozen ground) and the Dashalong station (in the transition zone). In the open area around the A'rou frozen ground superstation, a pair of Large Aperture Scintillometers (LASs) was installed to measure the sensible heat flux of the land surface. Meteorological data were generally logged every 30 min and can be aggregated to hourly, monthly and yearly values per request (Liu *et al.*, 2011). Steps of the AMS data processing and quality control were two-fold: (1) All the AWS data were averaged over an interval of 30 min for a total of 48 records per day. The missing data were denoted by -6999; (2) The un-physical data were rejected, and the gaps were denoted by -6999.

### 3.2.1 Air temperature and humidity

Air temperature and humidity were monitored at a height of 5 m above the ground every 30 min during various periods at all nine stations. At the A'rou frozen ground superstation, air temperature and humidity were observed at heights of 1, 2, 10, 15 and 25 m every 30 mins from 14 October 2012 to 31 December 2017. Consistent trends for both variables were noticed at all heights. However, air humidity showed a decreasing pattern with increasing heights. At locations other than the Yakou snow superstation, surface temperature was observed every 30 min from 24 June 2014 to 4 July 2014 using two sets of equipment for cross-comparison. Almost identical data were obtained between the two sets, although with variations among sites.

### 3.2.2 Radiation

Radiation data were collected every 30 min during various periods by four-component radiometers installed at all the AMSs (at a height of 6 m at seven standard AMSs and at a height of 1 m at the Yakou snow superstation), which include downward shortwave radiation (DSR, solar radiation), upward shortwave radiation (USR), downward longwave radiation (DLR), upward longwave radiation (ULR), and net radiation (Rn). In general, the DSR, DLR, ULR and Rn data were consistent among sites and throughout the years but varied with seasonal changes. However, the USR data exhibited significant differences among sites, specifically at the A'rou south-facing slope station and the Yakou snow superstation. At the Yakou snow superstation, the USR data were noticeably higher than the other stations due to the high albedo of snow cover. In contrast, minimum USR was found at the south-facing slope possibly due to sufficient sunshine and little snow cover distributed in the area. At the

A'rou frozen ground superstation, to observe vegetation in a typical alpine ecosystem, photosynthetically active radiation (PAR) was also monitored at an interval of 30 min during the observation period.

### 3.2.3 Wind speed and direction

Wind speed and direction were monitored 10 m above the ground at the interval of 30 mins during various periods. Specifically, at the A'rou frozen ground superstation, wind speed and direction were observed at heights of 1, 2, 5, 15 and 25 m every 30 mins from 14 October 2012 to 31 December 2017. Consistent data for both variables were noticed at all heights with variations among sites.

### 3.2.4 Precipitation and evapotranspiration (ET)

Among all the long-term monitored meteorological variables, precipitation and ET data at the Yakou, A'rou and Dashalong stations were collected and displayed in Figure 3. Precipitation data were measured by rain gauges, while ET rates were measured using the EC at the Dashalong, Yakou, and A'rou stations (Table 2). In particular, only the precipitation gauge (T200B, Geonor, USA) at the Yakou snow superstation was sheltered with DFIRs to collect both solid and liquid precipitation data. Because the uncertainties of the precipitation gauge (T200B) may result from the unstable voltage or unknown abnormity, evaporation of the liquid surface, and offset of the instrument, the postprocessing included three steps: (1) manual calibration by adding a certain amount of water into the gauge, (2) abnormal data rejection using the forward-backward filtering (Gustafsson, 1996), and (3) hourly precipitation calculation (using accumulated data before and after each hour). At the other stations, precipitation gauges (TE525M, Texas Electronics, USA) were neither sheltered by Alter shields nor DFIRs. Therefore, only liquid precipitation data were collected. Precipitation data were provided in raw format without any post-processing, which might be underestimated because of the wind and snowfall.

On the other hand, the instruments of EC were calibrated every six months, and the raw data acquired at 10 Hz were processed using the EdiRe software (University of Edinburgh, https://www.geos.ed.ac.uk/homes/jbm/micromet/EdiRe/), including spike detection and removal, lag correction of $H_2O/CO_2$ relative to the vertical wind component, sonic virtual temperature correction, coordinate rotation (2-D rotation), corrections for density fluctuation (Webb-Pearman-Leuning correction), and frequency response correction (Liu *et al*., 2011). EC data were subsequently averaged at an interval of 30 min and divided into three classes according to the quality assessment method of stationarity ($\Delta st$) and the integral turbulent characteristics test (ITC), as proposed by Foken and Wichura (1996): class 1 (level 0: $\Delta st<30$ and ITC<30), class 2 (level 1: $\Delta st<100$ and ITC<100), and class 3 (level 2: $\Delta st>100$ and ITC>100), which represent high-, medium-, and low-quality data, respectively. In addition to the above processing steps, half-hourly flux data were screened using a four-step procedure: (1) data from periods of sensor malfunction were rejected; (2) data collected before or after 1 hour of precipitation were rejected; (3) incomplete 30 min data were rejected when the missing data constituted more than 3% of the 30 min raw record; and (4) data were rejected at night when the friction velocity ($u^*$) was less than 0.1 m/s (Blanken *et al*., 1998). There were 48 records per day, with gaps denoted by -6999.

The annual precipitation was approximately 570 mm and 420 mm at the A'rou frozen ground superstation and Dashalong station, respectively, while it reached 600-800 mm at the Yakou snow superstation. It is interesting to observe that unlike the other two stations, at the Yakou snow superstation, a remarkable amount of ET was found not only in summer but also in spring and fall due to excessive ET on the surface of the snow cover. Also shown in Figure 3, similar patterns were clearly observed between precipitation and ET with possible correlations that could be investigated further through hydrological modeling and data analysis.

### 3.3 Snow data

A comprehensive snow dataset from the Yakou snow superstation has been updated since the summer of 2013. The observed snow variables included snow depth, snow water equivalent (SWE) and blowing (drifting) snow. Data obtained from 1 January 2014 to 31 December 2017 are summarized in Figure 4, with additional plots for precipitation, ET, soil temperature and moisture. Precipitation and ET (Figure 4a) were explained in Sec. 3.2.4, while soil temperature (measured at six depths below the ground) and moisture (measured at three depths below the ground) under freeze-thaw cycles are presented in Figure 4(e-f) as they relate to precipitation and ET. More data with further analysis of frozen ground observations at the study site will be introduced in the next section. Snow density and the liquid water contents of the snowpack were also measured by a Snow Pack Analyzer (SPA, Sommer, Austria). Unfortunately, this did not work well due to the influence of strong wind. Therefore, snow density data, which can be calculated using snow depth and SWE data, are not available at present.

### 3.3.1 Snow depth

Snow depth was measured by SR50A (Campbell Scientific, USA), which determines the distance between the sensor and the target by sending out ultrasonic pulses and listening for the returning echoes reflected from the target. The original snow depth data were available from 1 January 2014 to 31 December 2017 (with 53 days missing due to power loss or instrument malfunction, marked with -6999 in the dataset) at an interval of 30 min. In postprocessing, ambient air temperature measured using WXT520 (Vaisala, USA) was used to calibrate the snow depth data (Ryan *et al.* 2008). Data were cross-compared with the measured SWE (introduced in the next subsection), suspicious values were deleted manually followed by noise filtering and, finally, data were averaged to daily output. As shown in Figure 4(b), the snow depth has two peaks in autumn and spring, respectively. From 2014 to 2017, the maximum snow depth mostly decreased (31 cm, 16 cm, 15 cm, and 28 cm, respectively), showing an apparent alteration annually. The largest snow depth was found in April, which indicates that in this region, spring snowfall is the dominant and most unique feature in the hydrologic cycle. The measured snow depth data fluctuated in a chaotic pattern due to the strong solar radiation and blowing snow locally, as well as the dry winter, which is typical at the study site.

### 3.3.2 Snow water equivalent (SWE)

Snow water equivalent (SWE) was measured by GMON (CS725 GammaMONitor, Campbell Scientific, USA) from 1 January 2014 to 31 December 2017 at a temporal resolution of 6 hr. GMON provides the SWE data by measuring the absorption of natural ground gamma radiation through the snow layer, which depends on the mass of water between the source (the ground) and the radiation detector. The measurement area of the GMON was 50-100 $m^2$. By tracking the number of hits recorded on a daily basis from potassium ($^{40}K$) and thallium ($^{208}Tl$) energy windows located at 1.46 MeV and 2.61 MeV, superior performance was achieved compared with the traditional snow pillow measurement. Specifically, SWE data from GMON were calibrated by snow depth and density manually-measured using snow ruler and shovel twice a day (in the mornings and afternoons) in the spring of 2014. To avoid random uncertainties during calibration, a 100 m * 100 m grid around the GMON was designed to measure snow depth at an interval of 10 m (100 measuring spots in the grid). Snow density were also manually-measured within the grid at 6 selected locations. The averaged snow depth and density were used to fit the coefficients required by the GMON. With the daily collected SWE data (Figure 4c), the on-site snow season was defined from October to May (of the next year) each year at the Yakou snow superstation with a maximum value of approximately 120 mm. The daily snow depth and SWE data in the observation period exhibited similar patterns and are thus closely correlated as seen when comparing Figure 4(b) and 4(c).

### 3.3.3 Snow albedo

Snow albedo was measured using four-component radiometers described in Sec. 3.2.2 and calculated from the ratio between upward and downward radiations at short waves and long waves. As shown in Figure 4(d), a useful measure of the effective

albedo of the snowpack has been provided, which can be reasonably cross-compared with the snow depth and SWE (Figure 4b-c). It should be noted that the four-component radiation data (provided in raw format) and the albedo data shown in Figure 4d were calculated by the downward and upward shortwave radiation during 10:00-17:30 (local time) in order to filter out values at high solar zenith angles in early mornings and evenings.

### 3.3.4 Blowing (drifting) snow

Blowing (drifting) snow is commonly observed in the Qilian mountains due to its high altitude and complex terrain. It is important to estimate the occurrence of blowing snow because it may cause redistribution of the snow cover and influence the sublimation of snow (Li *et al*., 2018a). FlowCapt (IAV, Switzerland) was used to measure the number of blowing snow grains at 3 heights (0-1 m, 1-2 m, and 2-3 m) every 10 mins. The FlowCapt sensor is based on vibro-acoustics, which can provide good qualitative and quantitative information about drifted snow intensity and duration of snowdrift periods. The blowing snow data were available from 1 January 2014 to 20 September 2016 without any gaps. To filter the wind noise during the observation (especially in summer), it was necessary to manually delete the suspicious data by comparing the results with the SWE and snow depth data. The data would be rejected when (1) snow depth was zero, (2) wind speed was less than 3 m/s, or (3) air temperature was higher than 10℃. The data collected in 2014 are plotted in Figure 5, which show that the blowing snow fluxes reached a maximum close to the land surface and decreased with height. Two extreme events were found in May and September during that year.

### 3.4 Frozen ground data

To observe the FTCs of the frozen ground, soil moisture and temperature were measured at all the stations below ground (Table 2). At seven standard AMSs and the Yakou snow superstation, soil temperature was observed at 8 depths (0, 4, 10, 20, 40, 80, 120, 160 cm), while soil moisture was observed at 7 depths (4, 10, 20, 40, 80, 120, 160 cm). In addition, intensive measurements of both variables in 18 nested layers within 3.2 m below ground were performed at the A'rou frozen ground superstation (Table 2, Figure 6a) to capture the water and energy exchange within the soil. In addition, soil water potential and soil thermal conductivity were measured in the top 6 layers at an interval of 30 min (Table 2, Figure 6a). The datasets are available from 5 December 2012 to 31 December 2017, with intermittent loss mostly due to power loss and sensor malfunctions at high altitudes. The frozen ground data were provided in raw format without any post-processing. Soil water potential and moisture data are presented in Figure 6(b-c), showing consistent trends in topsoil layers if cross-compared, because the soil water potential represents the capacity for storage of moisture within the soil. Figures 6(b-c) also show that, during the melting season, relatively low soil moisture data were recorded due to the frozen status of the soil, while a sharp increase and fluctuations of the moisture data were noticed until spring in most of the layers. It was also found that soil moisture was quite low without clear seasonal variations in deeper soil layers (>60 cm), which may be because it is difficult for time domain reflectometry (TDR) sensors to work in bedrock. Figure 6(d) plots the soil heat conductivity in selected layers (10, 40 and 80 cm). Soil heat conductivity is difficult to analyze because it is a function of soil density, soil moisture content (ice content in frozen ground) and soil temperature, which cannot be easily calculated using a simple formula. Therefore, site observations can be utilized to evaluate the parameterization of alpine hydrologic models. The soil temperature changes within different layers under the FTCs, including the spatial and temporal variations, are clearly shown in Figure 6(e). Correlated patterns were found between soil moisture and temperature profiles when comparing Figure 6(c) and (e), variations of which become more significant close to the land surface. In addition, at the Yakou snow superstation, the soil at 160 cm below ground was mostly frozen before 2016, with a significant increase to 0℃ after 2016, which indicates that the active layer thickness has since increased.

## 4. Conclusions

In this paper, a suite of alpine hydrology datasets observed from an integrated, distributed and multiscale hydrometeorological – snow – frozen ground observation network in the upper reaches of the Heihe River Basin (HRB) is presented. With recent observational advances and decades of efforts, the integrated network has served as a testbed for alpine hydrology, cryospheric and meteorological sciences. The associated open-access datasets include high-quality hydrometeorological data with a focus on snow and frozen ground observations at the sub-basin and basin scales, which aims to address a variety of scientific questions including but not limited to: (1) how to provide accurate and effective forcing data for alpine hydrological models, (2) how to integrate observation, simulation and the acquired datasets for informative understanding, and (3) how perturbations (climate change and human activities) alter hydrologic processes. Our continuous efforts based on the current observation network are three-fold: (1) connecting "below-to-above" ground observations (e.g., incorporating other datasets from airborne and geophysical measurements) and looking into a more systematic investigation of the watershed behavior in an unprecedented manner; (2) developing robust data management assimilation tools and, ultimately with the aid of high performance computing, building a data-driven modeling platform to accelerate new discoveries and insights; and (3) performing outreach and offering rich possibilities for community collaboration, both within and across sites.

### Author contributions

TC, XL, SL, HL, RJ, MM, and JW designed the observations, TC, HL, ZX, JT, LX, JD and XY cleaned and organized the datasets. ZX, JT, YZ and ZR maintained the observation network. TC, XY and XL drafted the manuscript, and all authors contributed to the paper writing.

### Competing interests

The authors declare that they have no conflict of interest.

### Special issue statement

This article is part of the special issues "Hydrometeorological data from mountain and alpine research catchments". It is not associated with a conference.

### Review statement

This paper was edited by John Pomeroy as the Topical Editor of the Special Issue and reviewed by two anonymous referees.

### Acknowledgements

This study was supported by the Strategic Priority Research Program of the Chinese Academy of Sciences (grant No. XDA19070101 and XDA20100104), and the National Natural Science Foundation of China (grant No. 41531174 and 41271356).

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

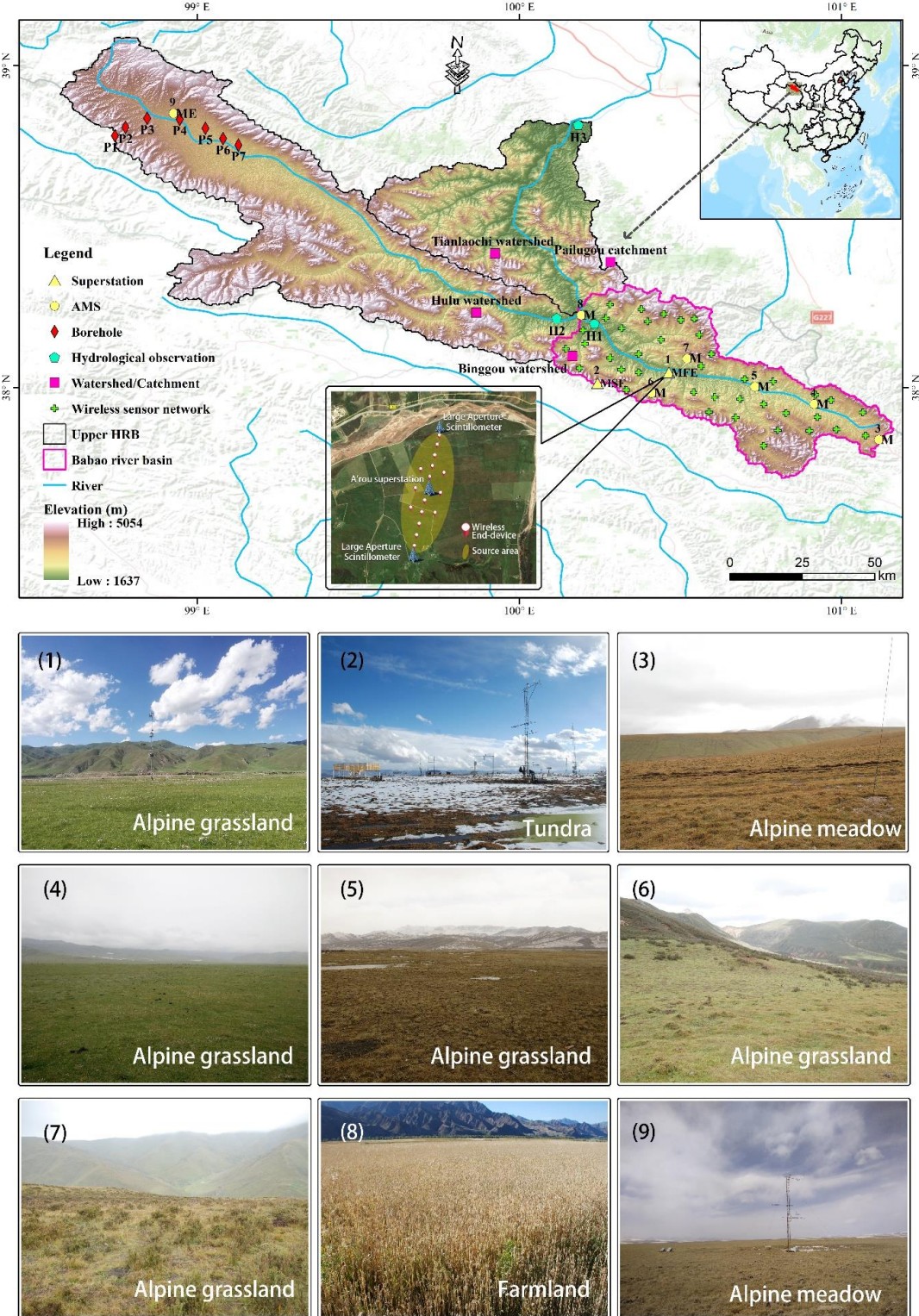

**Figure 1: The integrated hydrometeorological – snow – frozen ground observation network located at the study site. The observation network includes seven meteorological stations (M), one frozen ground superstation (F), and one snow superstation (S). The frozen ground and snow superstation observe the meteorological variables, evapotranspiration (E), and water/carbon fluxes, while the Dashalong station (No. 9) observes evapotranspiration by eddy covariance. The other observation watersheds (purple square) in the upper HRB include the Hulu watershed (Chen *et al*., 2014), Pailugou watershed (He *et al*., 2012), Tianlaochi watershed (Peng *et al*., 2014), and Binggou watershed (Li and Wang, 2011). There are nine boreholes in the west for permafrost observation (red diamond, Peng *et al*., 2016). In addition, 40 wireless sensor nodes (end-device, green cross) were designed to observe soil moisture and temperature in a sub-basin and the Babao River Basin (bounded by purple polygons, more details and data can be found in Jin *et al*., 2014). Additionally, there are three hydrological sites in the upper HRB where the runoff was measured (blue circle). In the zoomed-in subfigure at the bottom, a pair of large aperture scintillometers (LASs) are shown on the north-facing and south-facing slopes of the A'rou frozen ground superstation, in the middle of which 16 wireless end-devices (yellow dots) were installed for soil moisture and temperature measurements. To illustrate the ecosystems in the study area, nine typical landscapes are presented as listed in Table 1.**

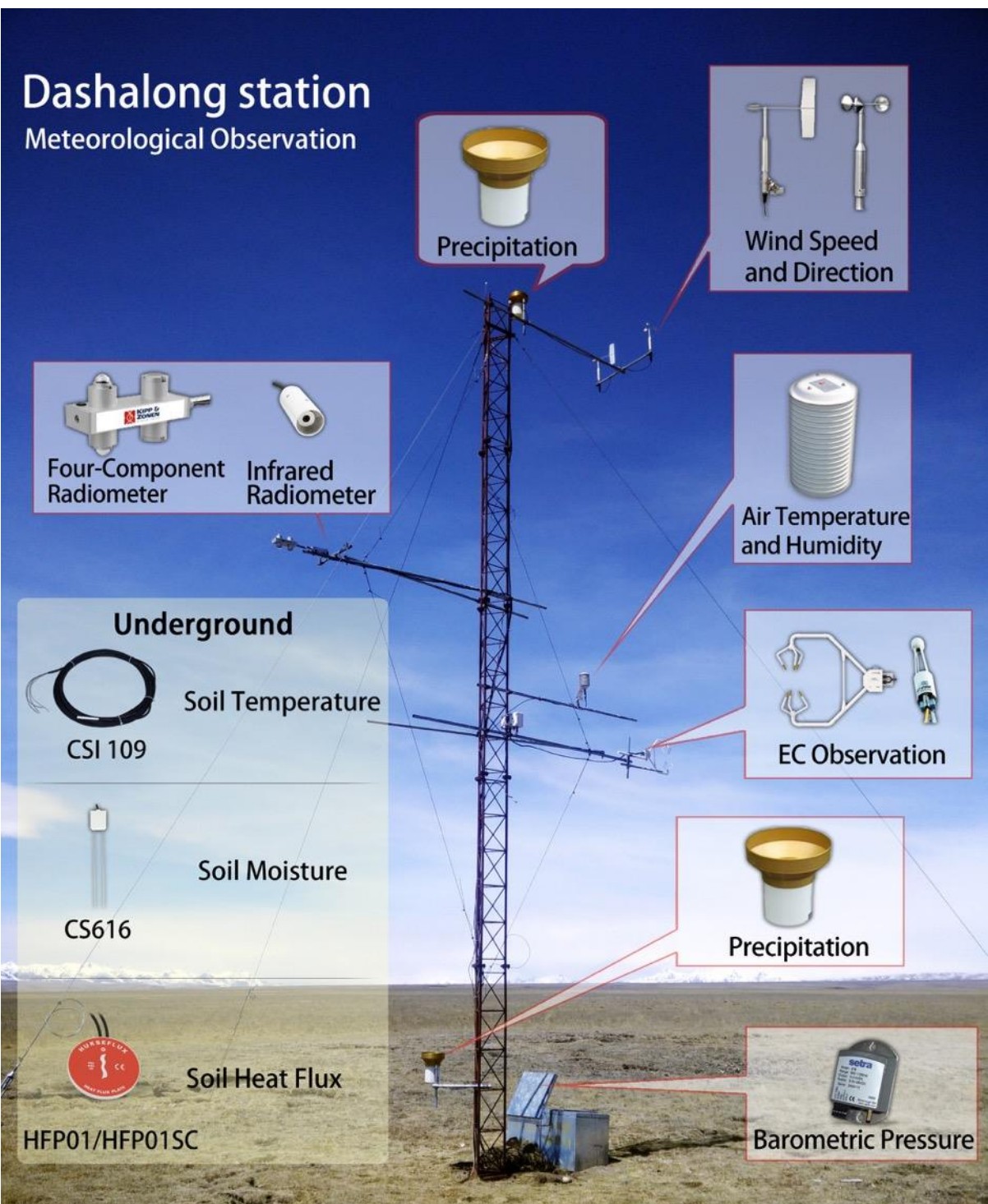

(a)

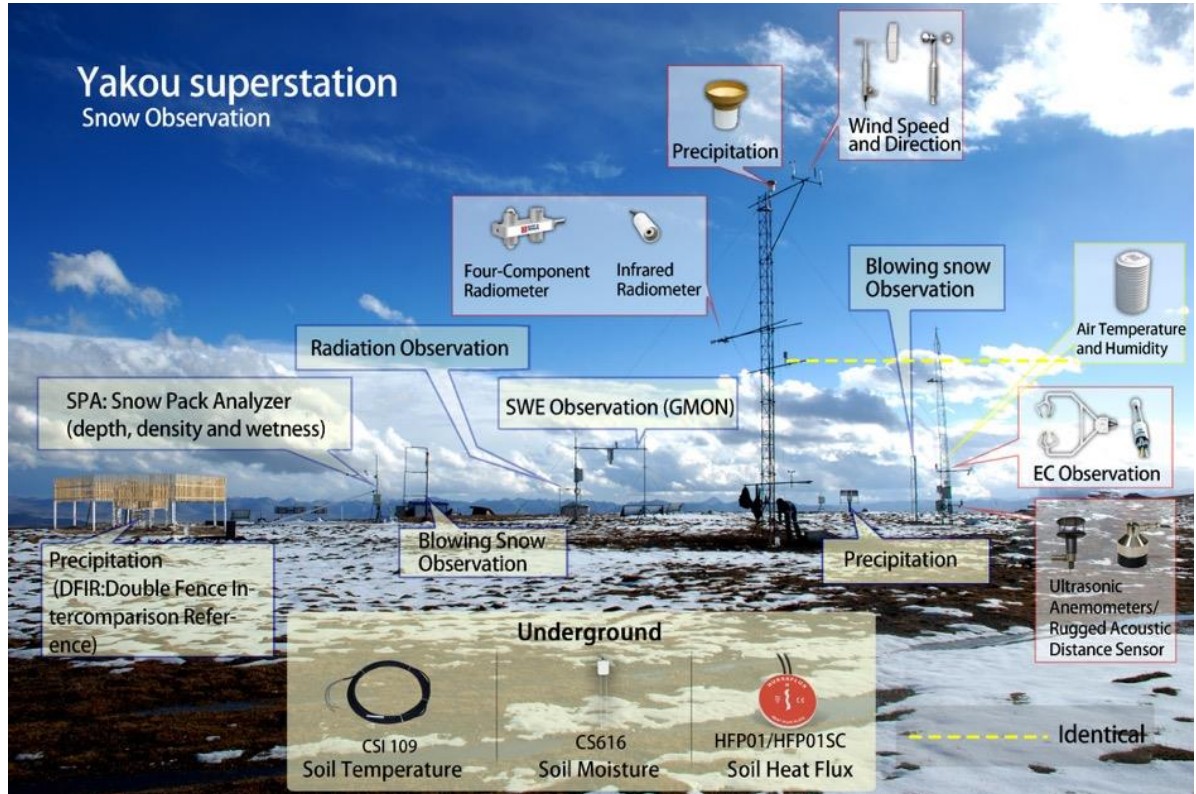

(b)

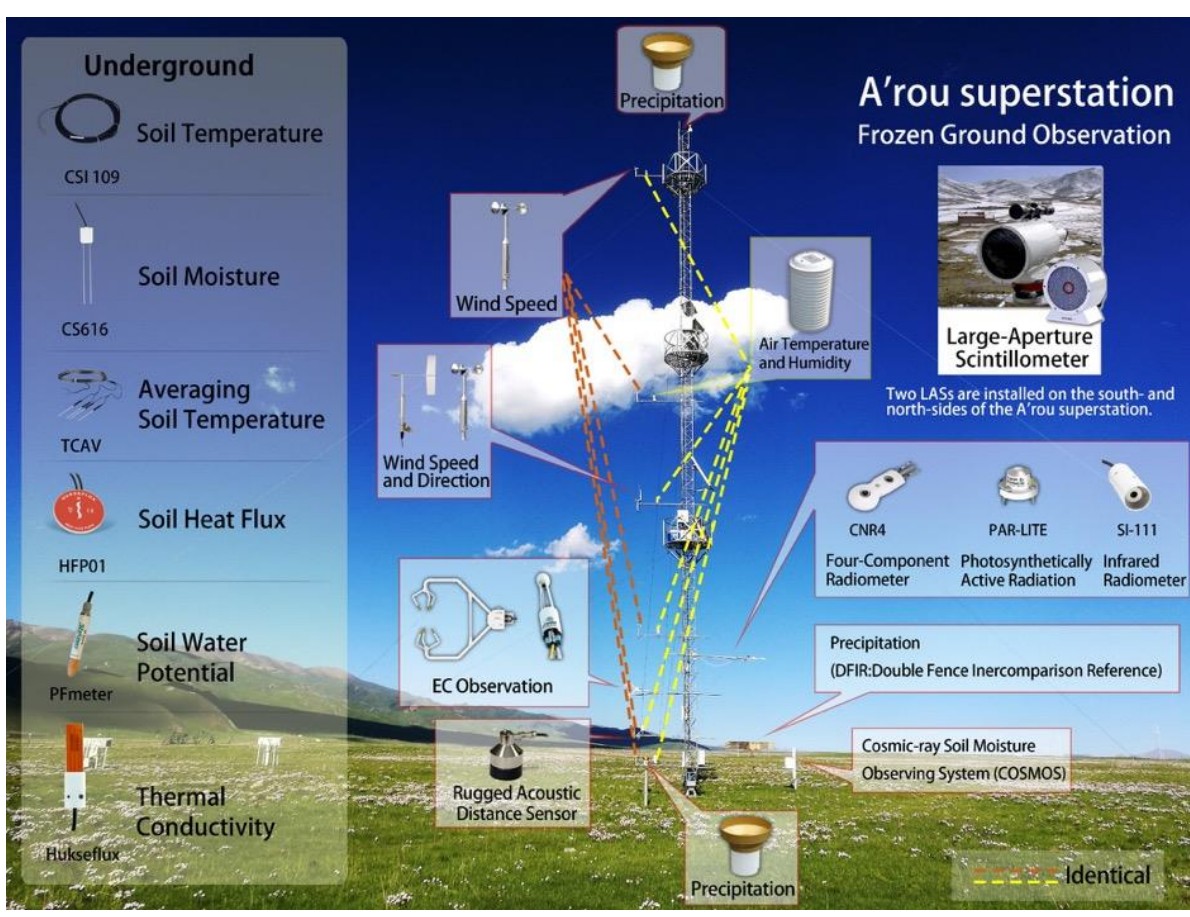

 (c)

**Figure 2: Experimental design: (a) hydrometeorological observation, (b) snow observation, and (c) frozen ground observation.**

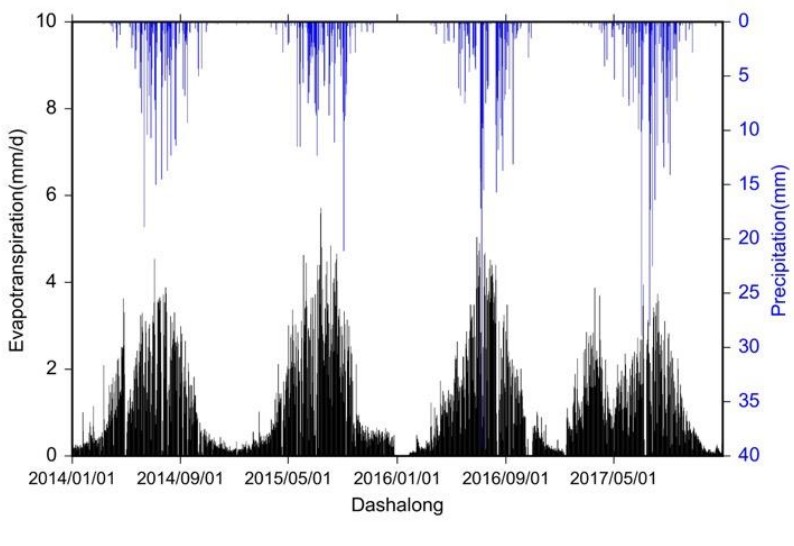

(a)

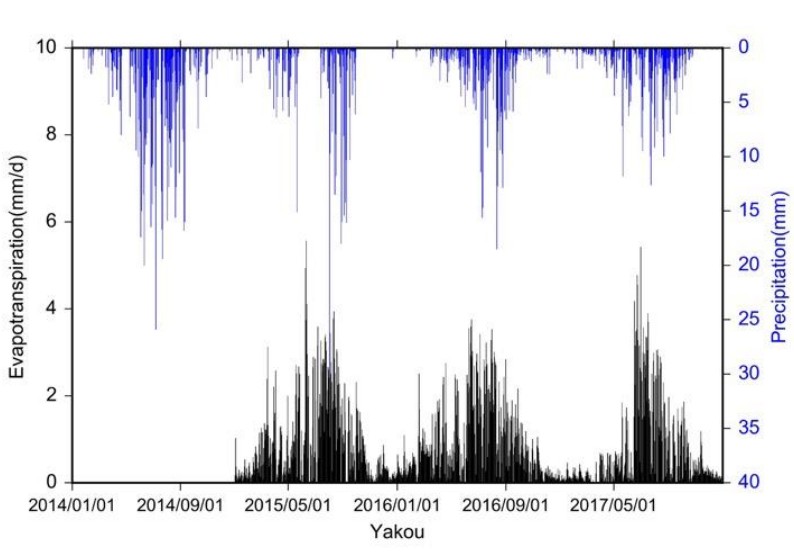

(b)

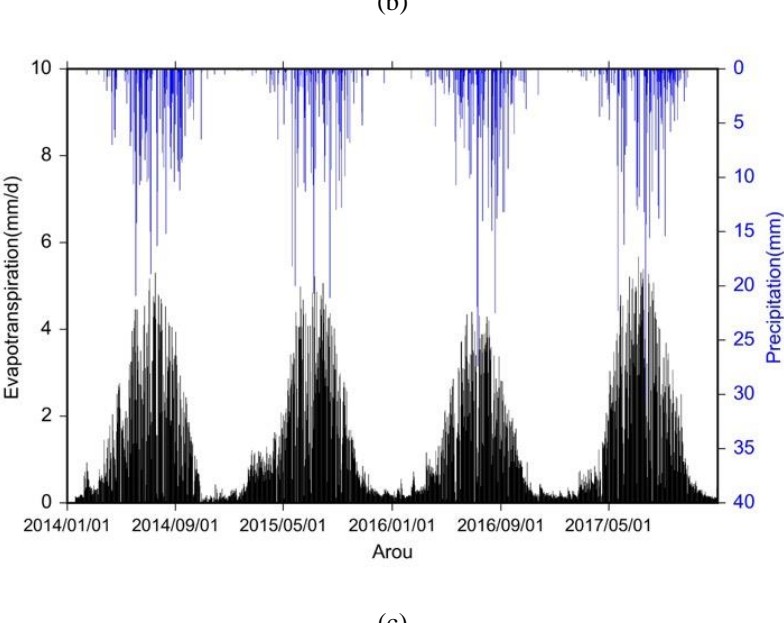

(c)

**Figure 3: Daily precipitation and ET data at the (a) Dashalong, (b) Yakou, and (c) A'rou stations from 2014 to 2017.**

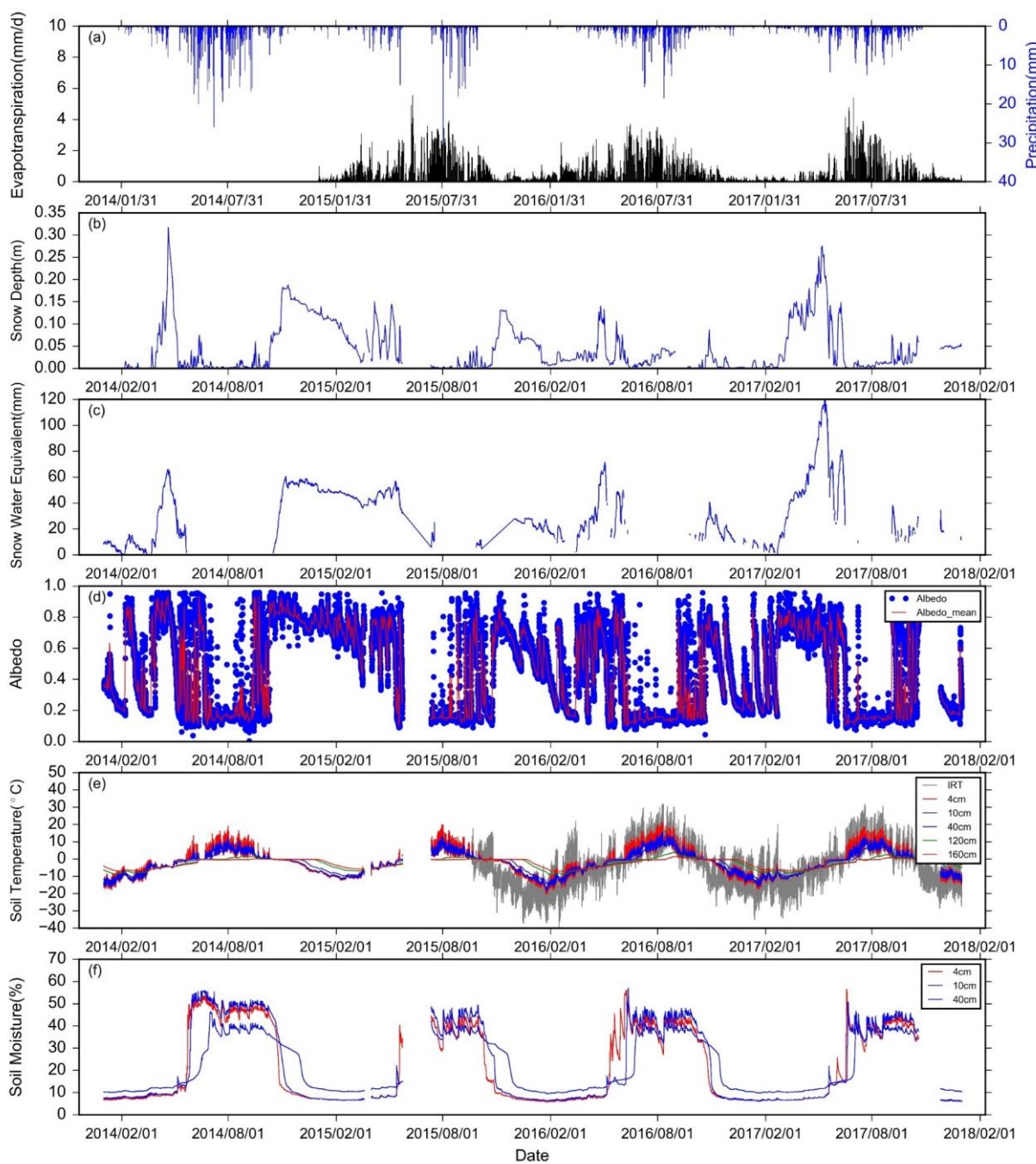

**Figure 4: Observations at the Yakou snow superstation from 2014/1/1 to 2017/12/31, including (a) precipitation and ET, (b) snow depth, (c) SWE, (d) albedo, (e) soil temperature, and (f) soil moisture.**

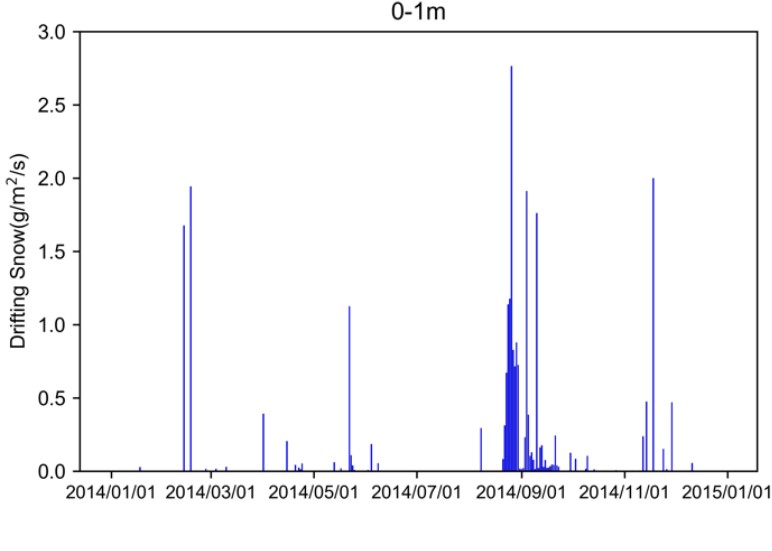

(a)

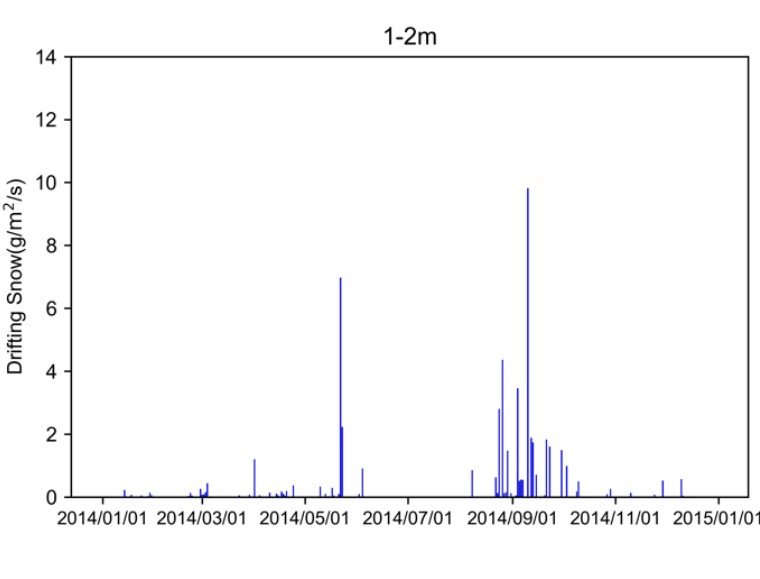

(b)

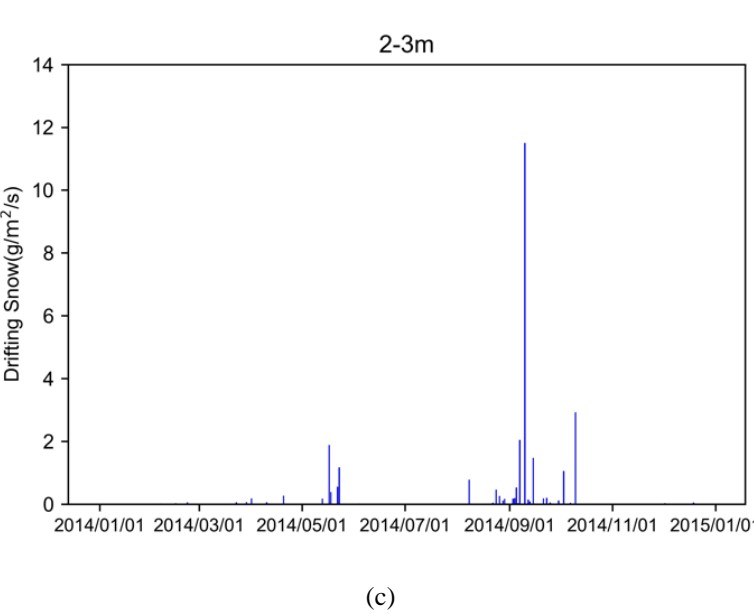

(c)

**Figure 5: Blowing snow measurement at the Yakou snow superstation from 2014/1/1 to 2014/12/31: (a) at the height of 0-1 m, (b) at the height of 1-2 m, and (c) at the height of 2-3 m.**

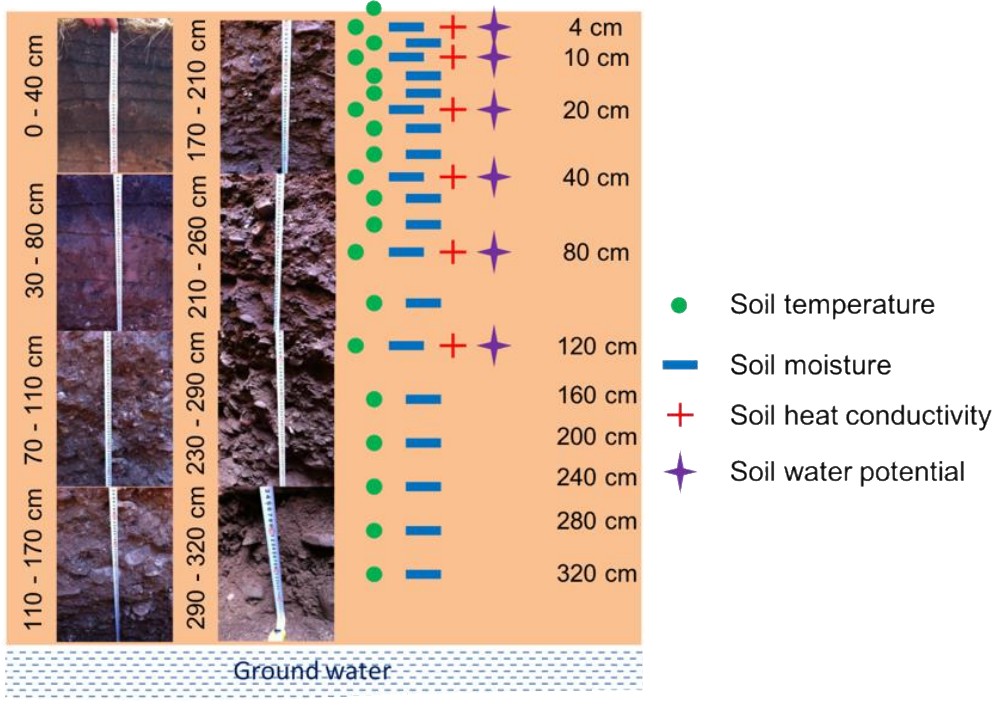

(a)

(b)

(c)

(d)

(e)

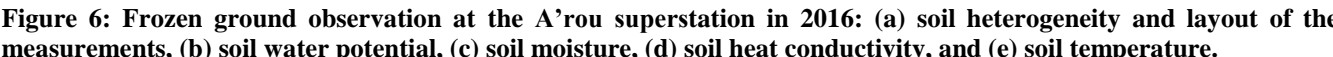

**Figure 6: Frozen ground observation at the A'rou superstation in 2016: (a) soil heterogeneity and layout of the measurements, (b) soil water potential, (c) soil moisture, (d) soil heat conductivity, and (e) soil temperature.**

**Table 1: List of seven AMSs and two superstations with detailed information.**

| ID | Name | Longitude | Latitude | Elevation (m) | Legend | Land cover | Observation period |
|----|------|-----------|----------|---------------|--------|------------|--------------------|
| 1 | A'rou frozen ground superstation*+ | 100.46°E | 38.05°N | 3033 | MFE | Alpine grassland | October 2012 - |
| 2 | Yakou snow superstation* | 100.24°E | 38.01°N | 4145 | MSE | Tundra | January 2014 - |
| 3 | Jingyangling station | 101.12°E | 37.84°N | 3750 | M | Alpine meadow | August 2013 - |
| 4 | E'bao station | 100.92°E | 37.95°N | 3294 | M | Alpine grassland | June 2013 -September 2016 |
| 5 | Huangcaogou station | 100.73°E | 38.00°N | 3137 | M | Alpine grassland | June 2013 – April 2015 |
| 6 | A'rou north-facing station | 100.41°E | 37.98°N | 3536 | M | Alpine grassland | August 2013 -December 2014 |
| 7 | A'rou south-facing station | 100.52°E | 38.09°N | 3529 | M | Alpine grassland | August 2013 -September 2015 |
| 8 | Huangzangsi station | 100.19°E | 38.23°N | 2612 | M | Farmland | June 2013 – April 2015 |
| 9 | Dashalong station* | 98.94°E | 38.84°N | 3739 | ME | Alpine meadow | August 2013 - |

Note: * indicates that the flux was observed by eddy covariance (EC). + indicates that the flux was observed by LAS. Legend: M, F, S, and E represent the hydrometeorological, frozen ground, snow cover and evapotranspiration observations.

**Table 2: Measuring variables, sensors and locations.**

| Variables | Sensors | Range | Accuracy | 7 AMSs | Yakou snow superstation | A'rou frozen ground superstation |
|---|---|---|---|---|---|---|
| Air temperature (Ta, ∘C) | Vaisala: HMP45C, HMP45AC | -40° to +60°C | ±0.2°C@20°C | 5 m | 5m | 1, 2, 5, 10, 15, and 25 m |
| | HMP45D, HMP45AD | -40° to +60°C | ±0.2°C@20°C | | | |
| Air humidity (RH, %) | Vaisala: HMP45C, HMP45AC | 0 to 100%RH | @ 20°C: ±2%RH(0 to 90%); ±3%RH(90% to 100%) | 5 m | 5m | 1, 2, 5, 10, 15, and 25 m |
| | HMP45D, HMP45AD | 0.8 to 100%RH | @ 20°C: ±2%RH(0 to 90%) ±3%RH(90% to 100% ) | | | |
| Wind speed (WS, m/s) | MetOne: 010C | 0 to 60m/s | ±0.07 m/s | 10 m | 10m | 1, 2, 5, 10, 15, and 25 m |
| | 034B | 0 to 75m/s | ±0.12 m/s for WS < 10.1 m/s; ±1.1% of reading for WS > 10.1 m/s | | | |
| | RM Young: 03001 | 0 to 50m/s | ±0.5 m/s | | | |
| Wind direction (WD, °) | MetOne: 020C | 0 to 360° | ±3° | 10 m | 10m | 2 m |
| | 034B | 0 to 360° | ±4° | | | |
| | RM Young: 03001 | 0 to 360° | ±5° | | | |
| Four-component radiation (DR/UR/DLR_ | Kipp&Zone: CNR1 | Pyranometer: 0 to 2000 W/m$^2$ Pyrgeometer: | Uncertainty in daily total: Pyranometer: ±10% Pyrgeometer ±10% | 6 m | 6m | 5 m |

| | | | | | | |
|---|---|---|---|---|---|---|
| Cor/ULR_Cor, $W/m^2$) | | -250 to 250 $W/m^2$ | | | | |
| | Kipp&Zone: CNR4 | Pyranometer: 0 to 2000 $W/m^2$ Pyrgeometer: -250 to 250 $W/m^2$ | Uncertainty in daily total: Pyranometer: < 5% Pyrgeometer:< 10% | | | |
| Photosynthetically active radiation (PAR, μmol/(s $m^2$)) | Kipp&Zone: PQS-1 | 0 to 10000 μmol/( $m^2$ s) | 4 to 10μv/μmol/( $m^2$ s) | | | 6m |
| | PAR-LITE | 0 to 10000 μmol/($m^2$ s) | 4 to 6μv/μmol/( $m^2$ s) | | | |
| Infrared temperature (IRT, ◦C) | Apogee: SI-111 | -40° to 70°C | ±0.2°C@ -10°C to +65°C; ±0.5°C@ -40°C to +70°C | 6 m (2 Repeats) | 6 m (2 Repeats) | 5m (2 Repeats) |
| | Avalon: IRTC3 | -20° to 60°C | ±0.3°C | | | |
| Precipitation (Rain, mm) | Texas Electronics: TE525M | 0° to +50°C | Up to 10mm/hr: ±1% 10 to 20mm/hr: +0, -3% 20 to 30mm/hr: +0, -5% | 10 m | 3m (DFIR), 10m | 5 m (DFIR) |
| | Geonor: T200BM3 | -40° to 60°C | 0.1% FS | | | |
| Air pressure (P, hpa) | Setra: CS100 | 600-1100hPa | ±0.5hPa at +20 °C | 0.5 m | 0.5m | 2 m |
| | Vaisala: PTB110 | 500-1100hPa | ±0.3 hPa at +20 °C | | | |
| Eddy covariance (EC) | Campbell Scientific Instrument (CSI), LI-COR: CSAT3 & Li7500A CSAT3 & Li7500 | $CO_2$: 0-3000ppm $H_2O$: 0-60ppt | $CO_2$: Within %1 of reading $H_2O$: Within 2% of reading | 4.5 m (Dashalong only) | 3m | 3.5 m |

| | | | | | | |
|---|---|---|---|---|---|---|
| Large Aperture Scintillometer (LAS) | SCINTEC: BLS450; Rainroot: ZZLAS | 250m - 6km | | | | 9.5 m |
| Soil heat flux (Gs, W/m²) | Hukseflux: HFP01SC | ±2000W/m² | ±3 of reading | 6 cm below ground (3 Repeats) | 6 cm below ground (3 Repeats) | 6 cm below ground (3 Repeats) |
| | HFP01 | ±2000W/m² | within -15% to +5% in 12hour totals | | | |
| | Avalon: HFT3 | ±100W/m² | <±5 of reading | | | |
| Average soil temperature (TCAV, °C) | Avalon: TCAV | -55-+85°C | ±0.3°C | | | 2 and 4cm |
| Soil temperature profile (Ts, °C) | CSI: 109, 109ss | -40° to +70°C | -40°C: ±0.6°C tolerance 0°C: ±0.38°C tolerance 25°C: ±0.1°C tolerance 50°C: ±0.3°C tolerance 70°C: ±0.4°C tolerance | 0, 4, 10, 20, 40, 80, 120, and 160 cm below ground | 0, 4, 10, 20, 40, 80, 120, and 160 cm below ground | 2, 4 (3 Repeats), 6, 10 (3 Repeats), 15, 20, 30, 40, 60, 80, 120, 160, 200, 240, 280, and 320 cm below ground |
| | Avalon: AV-10T | -45° to +65°C | <±0.2°C over 0°C to 60°C; ±0.4 @ -35°C | | | |
| Soil moisture profile (Ms, %) | CSI: CS616 | 0% to 50% VWC | ±2.5% VWC using standard calibration with bulk EC of ≤0.5 dS m-1, bulk density of ≤1.55 g cm-3) | 4, 10, 20, 40, 80, 120, and 160 cm below ground | 4, 10, 20, 40, 80, 120, and 160 cm below ground | 2, 4 (3 Repeats), 6, 10 (3 Repeats), 15, 20, 30, 40, 60, 80, 120, 160, 200, 240, 280, and 320 cm below ground |
| | Decagon: ECH2O-5 | 0% to 100% VWC | ±0.03 m³/m³ typical in mineral soils that have solution EC <8 dS/m; Medium specific calibration: ±0.02 m³/m³ in any porous medium (± 2%) | | | |
| Soil water potential (soil_pf, pF) | GeoPrecision: pF-meter | pF: 0-7 | ± 0.05 | | | 4, 10, 20, 40, 80, and 120 cm below ground |

| | | | | | | |
|---|---|---|---|---|---|---|
| Soil thermal conductivity (Soil_TCon, W/(m·K)) | Hukseflux: TP01 | 0.3 to 5 W/(m•K) | ± 5% | | | 4, 10, 20, 40, 80, and 120 cm below ground |
| Snow depth (mm) | CSI: SR50A | 0-10m | ±1cm | | 2.5 m | 2m |
| Snow water equivalent (SWE, mm) | CSI: CS725 | 0-600mm | ±15 mm (from 0 to 300 mm) ±15% (from 300 to 600 mm) | | 2.5m | |
| Drifting snow (snowdrift, g/m$^2$/s) | IAV: FlowCapt | 0-250 g/m$^2$/s | 1 g/m2/s | | 0-1m, 1-2m, and 2-3m | |

575

Note (DFIR): T200BM3 with Double Fence Intercomparison Reference (DFIR). The DFIRs were established in October 2016 at the Yakou snow superstation, and in August 2017 at the A'rou frozen ground superstation, respectively.