# Peer review of "Integrated hydrometeorological – snow – frozen ground observations in the alpine region of the Heihe River Basin, China"

_Earth System Science Data, 2019_

## Referee Comment (RC1) · Anonymous Referee #1 · 23 Mar 2019

Summary: This communication provides an interesting and highly suitable contribution to Earth System Science Data (ESSD). The paper describes an extensive hydrometeorological dataset collected in the upper reaches of the Heihe River Basin in China since the early 2010s. The paper is generally well-written and figures are very clear and entirely appropriate to illustrate key aspects of the dataset. This report provides guidance that the authors should consider in revising their manuscript.

General Comments:

1) One requirement for publication in ESSD is the inclusion of clear statements on the limitations of the datasets, which are lacking in the paper. For instance, what are

the operating range, accuracy and precision of the instrumentation used? Are there gaps in the datasets and was in-filling performed on these gaps (if any)? Was there any quality control/analysis performed on the data? In any case, information on the limitations of the observational data should be included in a revised paper.

2) As one example of possible data limitations, some of the precipitation gauges were not sheltered by Alter shields or DFIRs. As such, were precipitation data corrected for wind undercatch or provided in raw format without corrections?

3) How unique is this dataset relative to other hydrometeorological networks in operation in the Heihe River Basin? Are these the only hydrometeorological stations in the study area or do they complement an existing array of stations?

4) The journal requires statements on the author contributions and competing interests, as well as a special issue statement, prior to the Acknowledgements. This information should be added in a revised version of the manuscript.

Specific and Technical Comments:

1) P. 1, line 29: Change to "cryospheric science".

2) P. 1, line 34: The terminology here should be modified here as "regulated" headwaters suggest a human influence or interference on water resources, which I do not think is the intent of the statement here.

3) P. 1, line 35: Replace "reformed" by "altered" or "modified".

4) P. 1, line 37: Revise "impact factors" to just "impacts".

5) P. 2, line 46: Delete "the" before "observations".

6) P. 2, line 55: Replace "manipulate" with "control" or "influence".

7) P. 2, line 76: Should the abbreviation for "Global Terrestrial Observing Network" be "GTON", and not "GTOS"?

8) P. 2, line 82: Pluralize the following words: "alpine meadows, forests, irrigated crops, riparian ecosystems, and deserts". At the end of this line, do you mean "distributed along an altitudinal gradient"?

9) P. 3, line 86: Replace "great" with "marked".

10) P. 3, line 87: Revise "the mountainous region" to "mountainous regions".

11) P. 3, line 112: Insert the units for the altitudes.

12) P. 3, line 116: Use a word other than "fascinating".

13) P. 3, line 119: Rather than "west wind circulation" do you mean "mid-latitude westerlies"?

14) P. 3, line 122: What is meant by "hydrothermal"?

15) P. 3, lines 122-123: Pluralize the words so that it reads "alpine grasslands (dominant), alpine shrubs, alpine meadows, tundra, deserts and forest steppes."

16) P. 3, line 124: Pluralize "grasses and forests".

17) p. 4, line 131: Replace "fast" with "rapid".

18) P. 4, line 152: Change to: "at each".

19) P. 4, line153: Change to "AMS installed at".

20) P. 5, line 168: Given the abbreviation "AMSs" has already been defined, use it here only and delete "Automatic Meteorological Stations".

21) P. 5, line 183: Replace "sophisticated" with "detailed".

22) P. 5, lines 195 and 197: Date format may need to change to that used by the journal.

23) P. 6, lines 213, 225, 226 and 236: Same comment.

24) P. 6, line 234: Should this read "Campbell Scientific"?

25) P. 6, line 242: Delete the attribution of recent changes in snow depth to climate change, this could just be due to interannual variability, changes in snow redistribution by wind, etc.

26) P. 7, lines 247-248 and 267: Date format may need to change.

27) P. 7, line 250: Write as "50-100 m2".

28) P. 7, line 262: Pluralize "Qilian Mountains".

29) P. 7, line 263: Change to "redistribution".

30) P. 7, line 269: Change to "which show that the blowing snow fluxes".

31) P. 8, line 283: Do you mean "during spring" or "after spring"?

32) P. 8, line 293: Again, do not attribute recent changes necessarily to climate change.

33) P. 8, line 297: Change to "cryospheric science".

34) P. 8, lines 298-299: Replace "with specific respect to" with "with a focus on".

35) P. 8, line 302: What is meant by "alter fundamental processes"?

36) P. 9, line 320: Change to "CALM".

37) p. 10, line 366: Change to "(PUB) – a review".

38) P. 11, line 406: Change to "Col de Porte, France" and correct the remainder of the title, it appears the units for elevation are missing.

39) p. 19, Table 1: Insert "(°E)" and "(°N)" for longitude and latitude, respectively. For the observation period, consider adding the starting and ending month as well.

---

## Referee Comment (RC2) · Anonymous Referee #2 · 29 Mar 2019

Summary: this manuscript presents a comprehensive dataset of hydrological variables above and below the ground surface at the Heihe River Basin, in China. The breadth of the data collection effort is commendable, and the dataset is potentially very suitable as a contribution to ESSD.

General Comments:

I agree with referee #1 in that a more thorough data quality assessment should be provided. If space is a concern, maybe an online supplementary material could be provided. Also, in a few instances it is mentioned that manual filtering was carried out before adopting a definitive dataset for a given variable. This is not unexpected, but

if no information about which individual data points correspond to filtered values, then it becomes problematic. Perhaps both a "raw" and a "postprocessed" data products should be presented.

A couple other questions about instruments and data (I focus on snow, as this is my area of expertise): in your figures, only TI rain gages are depicted. I imagine that the Geonor instruments are those located inside the DFIR setups? The TI's are not expected to measure solid precipitation properly, but the Geonors are. However, your data plots show zero or close to zero precip in winter, at the same time when snow depth and water equivalent are positive. Must we conclude that your stations are unable to record solid precipitation?

Then, you talk about snow data, and state that depth and SWE were obtained from the SR50 and the CS725 sensors, respectively. The SPA did not work, apparently, and you link this malfunction to wind and conclude that for this reason snow density is unavailable. However, you do measure depth and SWE with the other sensors! On any case, as this is a dataset paper, I would not expect it to present estimated or derived information (such as density), but only measured data. Additionally, did you make manual depth and SWE measurements with snow probes, samplers or pits? do these match what was recorded by the sensors?

Finally, albedo data looks good, but a bit noisy. Please mention at what solar angle ranges was albedo recorded. Did you filter out values at high angles in the early morning and evening?

Specific comments:

L55. Replace "manipulate". Maybe "drive" or "modulate" would be better.

L83: delete "the" before "altitude".

L85: delete "the" before "alpine".

---

## Author Comment (AC1) · 21 May 2019

**Summary:** This communication provides an interesting and highly suitable contribution to Earth System Science Data (ESSD). The paper describes an extensive hydrometeorological dataset collected in the upper reaches of the Heihe River Basin in China since the early 2010s. The paper is generally well-written and figures are very clear and entirely appropriate to illustrate key aspects of the dataset. This report provides guidance that the authors should consider in revising their manuscript.

**Response:** We thank Anonymous Referee #1 for her/his positive feedback and insightful comments, which provided tremendous help for improving our manuscript. We have carefully addressed all the issues raised by the referee and modified our manuscript accordingly. Detailed responses (marked in blue font) are summarized in the following sections with the original comments (marked in black font). A clean version of the revised manuscript is also attached with changes marked in red font.

**General Comments:**

1) One requirement for publication in ESSD is the inclusion of clear statements on the limitations of the datasets, which are lacking in the paper. For instance, what are the operating range, accuracy and precision of the instrumentation used? Are there gaps in the datasets and was in-filling performed on these gaps (if any)? Was there any quality control/analysis performed on the data? In any case, information on the limitations of the observational data should be included in a revised paper.

**Response:** Points well-taken. We agree with the referee that it is important to provide detailed info of the instruments and the datasets (such as quality control and the limitations). In the revised manuscript, we have added the associated descriptions.

a.  The operating range, accuracy and precision of the instrumentation used in our observation network have been added in Table 2.

b.  In terms of data post-processing and quality control, we also provided more descriptions for each observation variable:

*Meteorological data*:

We used general post-processing and quality control for the meteorological data, the steps of which were stated in Sec. 3.2 (Page 5, Line 187-189) as:

"Steps of the AMS data processing and quality control were two-fold: (1) All the AWS data were averaged over an interval of 30 min for a total of 48 records per day. The missing data were denoted by -6999; (2) The un-physical data were rejected, and the gaps were denoted by -6999."

Precipitation data were further calibrated and post-processed as stated in Sec. 3.2.4 (Page 6, Line 216-224):

"In particular, only the precipitation gauge (T200B, Geonor, USA) at the Yakou snow superstation was sheltered with DFIRs to collect both solid and liquid precipitation data. Because the uncertainties of the precipitation gauge (T200B) may result from the unstable voltage or unknown abnormity, evaporation of the liquid surface, and offset of the instrument, the postprocessing included three steps: (1) manual calibration by adding a certain amount of water into the gauge, (2) abnormal data rejection using the forward-backward filtering (Gustafsson, 1996), and (3) hourly precipitation calculation (using accumulated data before and after each hour). At the other stations, precipitation gauges (TE525M, Texas Electronics, USA) were neither sheltered by Alter shields nor DFIRs. Therefore, only liquid precipitation data were collected. Precipitation data were provided in raw format without any post-processing, which might be underestimated because of the wind and snowfall."

Specifically, for the EC data, gap-filling was processed with quality control. Detailed descriptions were added in Sec. 3.2.4 (Page 6, Line 225-237) as:

"On the other hand, the instruments of EC were calibrated every six months, and the raw data acquired at 10 Hz were processed using the EdiRe software (University of Edinburgh,

https://www.geos.ed.ac.uk/homes/jbm/micromet/EdiRe/), including spike detection and removal, lag correction of $H_2O/CO_2$ relative to the vertical wind component, sonic virtual temperature correction, coordinate rotation (2-D rotation), corrections for density fluctuation (Webb-Pearman-Leuning correction), and frequency response correction (Liu *et al*., 2011). EC data were subsequently averaged at an interval of 30 min and divided into three classes according to the quality assessment method of stationarity (Δst) and the integral turbulent characteristics test (ITC), as proposed by Foken and Wichura (1996): class 1 (level 0: Δst<30 and ITC<30), class 2 (level 1: Δst<100 and ITC<100), and class 3 (level 2: Δst>100 and ITC>100), which represent high-, medium-, and low-quality data, respectively. In addition to the above processing steps, half-hourly flux data were screened using a four-step procedure: (1) data from periods of sensor malfunction were rejected; (2) data collected before or after 1 hr of precipitation were rejected; (3) incomplete 30 min data were rejected when the missing data constituted more than 3% of the 30 min raw record; and (4) data were rejected at night when the friction velocity ($u^*$) was less than 0.1 m/s (Blanken *et al*., 1998). There were 48 records per day, with gaps denoted by -6999."

*Snow data*:

*Snow depth* (Sec. 3.3.1, Page 7, Line 258-261):

"In postprocessing, ambient air temperature measured using WXT520 (Vaisala, USA) was used to calibrate the snow depth data (Ryan *et al*. 2008). Data were cross-compared with the measured SWE (introduced in the next subsection), suspicious values were deleted manually followed by noise filtering and, finally, data were averaged to daily output."

*Snow water equivalent* (SWE, Sec. 3.3.2, Page 7, Line 272-276):

"Specifically, SWE data from GMON were calibrated by snow depth and density manually-measured using snow ruler and shovel twice a day (in the mornings and afternoons) in the spring of 2014. To avoid random uncertainties during calibration, a 100 m * 100 m grid around the GMON was designed to measure snow depth at an interval of 10 m (100 measuring spots in the grid). Snow density were also manually-measured within the grid at 6

selected locations.  The averaged snow depth and density were used to fit the coefficients required by the GMON."

*Snow albedo* (Sec. 3.3.3, Page 8, Line 285-287):

"It should be noted that the four-component radiation data (provided in raw format) and the albedo data shown in Figure 4d were calculated by the downward and upward shortwave radiation during 10:00-17:30 (local time) in order to filter out values at high solar zenith angles in early mornings and evenings."

Blowing snow (Sec. 3.3.4, Page 8, Line 296-297):

"To filter the wind noise during the observation (especially in summer), it was necessary to manually delete the suspicious data by comparing the results with the SWE and snow depth data. The data would be rejected when (1) snow depth was zero, (2) wind speed was less than 3 m/s, or (3) air temperature was higher than 10°C."

*Frozen ground data* (Sec. 3.4, Page 8, Line 308):

"The frozen ground data were provided in raw format without any post-processing."

2) As one example of possible data limitations, some of the precipitation gauges were not sheltered by Alter shields or DFIRs. As such, were precipitation data corrected for wind undercatch or provided in raw format without corrections?

**Response:** Good point. We should have clarified the data/instrument limitations. Other than those explained in the response to the first question, we specifically added statements (limitations/uncertainties of the instrument/datasets, post-processing) for precipitation data in Sec. 3.2.4 (Page 6, Line 216-224) as:

"In particular, only the precipitation gauge (T200B, Geonor, USA) at the Yakou snow superstation was sheltered with DFIRs to collect both solid and liquid precipitation data. Because the uncertainties of the precipitation gauge (T200B) may result from the unstable voltage or unknown abnormity, evaporation of the liquid surface, and offset of the instrument, the postprocessing included three steps: (1) manual calibration by adding a certain amount of water into the gauge, (2) abnormal data rejection using the forward-backward filtering (Gustafsson, 1996), and (3) hourly precipitation calculation

(using accumulated data before and after each hour). At the other stations, precipitation gauges (TE525M, Texas Electronics, USA) were neither sheltered by Alter shields nor DFIRs. Therefore, only liquid precipitation data were collected. Precipitation data were provided in raw format without any post-processing, which might be underestimated because of the wind and snowfall."

3) How unique is this dataset relative to other hydrometeorological networks in operation in the Heihe River Basin? Are these the only hydrometeorological stations in the study area or do they complement an existing array of stations?

**Response:** In the upper reaches of the Heihe River Basin (HRB), there are other stations established by four research teams as mentioned in the Introduction (Page 3, Line 88-90) and shown in Figure 1. We pointed out that those stations were either located in small-catchments or with specific research area. On the other hand, there also exist other hydrometeorological stations managed by local meteorological agencies for various purposes such as weather forecast, not specifically for scientific research and not related to the current datasets. The automatic meteorological stations (AMSs) introduced in this study were installed/operated within the framework of China's first basin-scale integrated observatory network (Li *et al.*, 2009; Li *et al.*, 2013; Li *et al.*, 2017; Liu *et al.*, 2018) supported by the National Natural Science Foundation of China (NSFC) (stated in Page 2, Line 79-83). In particular, to investigate the alpine hydrology and cryospheric science in the upper reaches of the HRB, the AMSs along with the snow and frozen ground stations were built since 2013 during the Heihe Watershed Allied Telemetry Experimental Research (HiWATER, Li *et al*., 2013). There have been more stations for other research purposes in the middle and lower reaches of the HRB (Liu *et al*., 2018) (stated in Page 3, Line 94-96). In summary, by far, the datasets introduced in the current study fully exhibit the characteristics of the alpine region in the HRB and represent the complete and sophisticated observation efforts invested since the last decade.

4) The journal requires statements on the author contributions and competing interests, as well as a special issue statement, prior to the Acknowledgements. This information should be added in a revised version of the manuscript.

**Response:** We added "Author contributions", "Competing interests", "Special issue statement" and "Review statement" prior to the Acknowledgement.

**Specific and Technical Comments:**

1) P. 1, line 29: Change to "cryospheric science".

**Response:** Corrected.

2) P. 1, line 34: The terminology here should be modified here as "regulated" headwaters suggest a human influence or interference on water resources, which I do not think is the intent of the statement here.

**Response:** We deleted "that need to be regulated" in the texts.

3) P. 1, line 35: Replace "reformed" by "altered" or "modified".

**Response:** We replaced "reformed" by "altered".

4) P. 1, line 37: Revise "impact factors" to just "impacts".

**Response:** Corrected.

5) P. 2, line 46: Delete "the" before "observations".

**Response:** Deleted.

6) P. 2, line 55: Replace "manipulate" with "control" or "influence".

**Response:** We replaced "manipulate" with "influence".

7) P. 2, line 76: Should the abbreviation for "Global Terrestrial Observing Network" be "GTON", and not "GTOS"?

**Response:** We apologized for the typo. It should be "Global Terrestrial Observing System (GTOS)".

8) P. 2, line 82: Pluralize the following words: "alpine meadows, forests, irrigated crops, riparian ecosystems, and deserts". At the end of this line, do you mean "distributed along an altitudinal gradient"?

**Response:** We pluralized the words as "alpine meadows, forests, irrigated crops, riparian ecosystems, and deserts" and changed the texts as "distributed along an altitudinal gradient" at the end of this sentence.

9) P. 3, line 86: Replace "great" with "marked".

**Response:** We replaced "great" with "marked".

10) P. 3, line 87: Revise "the mountainous region" to "mountainous regions".

**Response:** Corrected.

11) P. 3, line 112: Insert the units for the altitudes.

**Response:** Corrected.

12) P. 3, line 116: Use a word other than "fascinating".

**Response:** We used "ideal" instead of "fascinating".

13) P. 3, line 119: Rather than "west wind circulation" do you mean "mid-latitude westerlies"?

**Response:** We used "mid-latitude westerlies" instead of "west wind circulation".

14) P. 3, line 122: What is meant by "hydrothermal"?

**Response:** We deleted "under dynamic hydrothermal conditions".

15) P. 3, lines 122-123: Pluralize the words so that it reads "alpine grasslands (dominant), alpine shrubs, alpine meadows, tundra, deserts and forest steppes."

**Response:** We pluralized the words as "alpine grasslands (dominant), alpine shrubs, alpine meadows, tundra, deserts and forest steppes".

16) P. 3, line 124: Pluralize "grasses and forests".

**Response:** We pluralized the words as "grasses and forests".

17) p. 4, line 131: Replace "fast" with "rapid".

**Response:** We replaced "fast" with "rapid".

18) P. 4, line 152: Change to: "at each".

**Response:** Corrected.

19) P. 4, line153: Change to "AMS installed at".

**Response:** Corrected.

20) P. 5, line 168: Given the abbreviation "AMSs" has already been defined, use it here only and delete "Automatic Meteorological Stations".

**Response:** Corrected.

21) P. 5, line 183: Replace "sophisticated" with "detailed".

**Response:** We replaced "sophisticated" with "detailed".

22) P. 5, lines 195 and 197: Date format may need to change to that used by the journal.

**Response:** Corrected. All the dates were changed to "DD Month YYYY" format.

23) P. 6, lines 213, 225, 226 and 236: Same comment.

**Response:** Corrected. All the dates were changed to "DD Month YYYY" format.

24) P. 6, line 234: Should this read "Campbell Scientific"?

**Response:** Corrected.

25) P. 6, line 242: Delete the attribution of recent changes in snow depth to climate change, this could just be due to interannual variability, changes in snow redistribution by wind, etc.

**Response:** Agreed. We delete "influenced by local climate".

26) P. 7, lines 247-248 and 267: Date format may need to change.

**Response:** Corrected. All the dates were changed to "DD Month YYYY" format.

27) P. 7, line 250: Write as "50-100 m2".

**Response:** Corrected.

28) P. 7, line 262: Pluralize "Qilian Mountains".

**Response:** Corrected.

29) P. 7, line 263: Change to "redistribution".

**Response:** Corrected.

30) P. 7, line 269: Change to "which show that the blowing snow fluxes".

**Response:** Corrected.

31) P. 8, line 283: Do you mean "during spring" or "after spring"?

**Response:** The "until spring" was changed to "during the melting seasons".

32) P. 8, line 293: Again, do not attribute recent changes necessarily to climate change.

**Response:** We deleted "due to climate warming".

33) P. 8, line 297: Change to "cryospheric science".

**Response:** Corrected.

34) P. 8, lines 298-299: Replace "with specific respect to" with "with a focus on".

**Response:** We replaced "with specific respect" with "with a focus on".

35) P. 8, line 302: What is meant by "alter fundamental processes"?

**Response:** We changed to "alter hydrologic processes".

36) P. 9, line 320: Change to "CALM".

**Response:** Corrected.

37) p. 10, line 366: Change to "(PUB) – a review".

**Response:** Corrected.

38) P. 11, line 406: Change to "Col de Porte, France" and correct the remainder of the title, it appears the units for elevation are missing.

**Response:** Corrected. It should be "Col de Porte, France, 1325 m altitude".

39) p. 19, Table 1: Insert "(°E)" and "(°N)" for longitude and latitude, respectively. For the observation period, consider adding the starting and ending month as well.

**Response:** "(°E)" and "(°N)" for longitudes and latitudes were added. Also, the starting and ending months of the observation periods were added.

---

## Author Comment (AC2) · 21 May 2019

**Summary:** this manuscript presents a comprehensive dataset of hydrological variables above and below the ground surface at the Heihe River Basin, in China. The breadth of the data collection effort is commendable, and the dataset is potentially very suitable as a contribution to ESSD.

**Response:** We thank Anonymous Referee #2 for her/his positive feedback and insightful comments, which provided tremendous help for improving our manuscript. We have carefully addressed all the issues raised by the referee and modified our manuscript accordingly. Detailed responses (marked in blue) are summarized in the following sections with the original comments (marked in black). A clean version of the revised manuscript is also attached with changes marked in red.

**General Comments:**

I agree with referee #1 in that a more thorough data quality assessment should be provided. If space is a concern, maybe an online supplementary material could be provided. Also, in a few instances it is mentioned that manual filtering was carried out before adopting a definitive dataset for a given variable. This is not unexpected, but if no information about which individual data points correspond to filtered values, then it becomes problematic. Perhaps both a "raw" and a "postprocessed" data products should be presented.

**Response:** We agree with both referees that detailed data quality assessment should have been provided. In the revised manuscript, we added data postprocessing (if there was any) and quality assessment for meteorological data, snow data and frozen ground data separately.

a. The operating range, accuracy and precision of the instrumentation used in our observation network have been added in Table 2.

b. In terms of data post-processing and quality control, we also provided more descriptions for each observation variable:

*Meteorological data*:

We used general post-processing and quality control for the meteorological data, the steps of which were stated in Sec. 3.2 (Page 5, Line 187-189) as:

"Steps of the AMS data processing and quality control were two-fold: (1) All the AWS data were averaged over an interval of 30 min for a total of 48 records per day. The missing data were denoted by -6999; (2) The un-physical data were rejected, and the gaps were denoted by -6999."

Precipitation data were further calibrated and post-processed as stated in Sec. 3.2.4 (Page 6, Line 216-224):

"In particular, only the precipitation gauge (T200B, Geonor, USA) at the Yakou snow superstation was sheltered with DFIRs to collect both solid and liquid precipitation data. Because the uncertainties of the precipitation gauge (T200B) may result from the unstable voltage or unknown abnormity, evaporation of the liquid surface, and offset of the instrument, the postprocessing included three steps: (1) manual calibration by adding a certain amount of water into the gauge, (2) abnormal data rejection using the forward-backward filtering (Gustafsson, 1996), and (3) hourly precipitation calculation (using accumulated data before and after each hour). At the other stations, precipitation gauges (TE525M, Texas Electronics, USA) were neither sheltered by Alter shields nor DFIRs. Therefore, only liquid precipitation data were collected. Precipitation data were provided in raw format without any post-processing, which might be underestimated because of the wind and snowfall."

Specifically, for the EC data, gap-filling was processed with quality control. Detailed descriptions were added in Sec. 3.2.4 (Page 6, Line 225-237) as:

"On the other hand, the instruments of EC were calibrated every six months, and the raw data acquired at 10 Hz were processed using the EdiRe software (University of Edinburgh,

https://www.geos.ed.ac.uk/homes/jbm/micromet/EdiRe/), including spike detection and removal, lag correction of $H_2O/CO_2$ relative to the vertical wind component, sonic virtual temperature correction, coordinate rotation (2-D rotation), corrections for density fluctuation (Webb-Pearman-Leuning correction), and frequency response correction (Liu *et al*., 2011). EC data were subsequently averaged at an interval of 30 min and divided into three classes according to the quality assessment method of stationarity ($\Delta$st) and the integral turbulent characteristics test (ITC), as proposed by Foken and Wichura (1996): class 1 (level 0: $\Delta$st<30 and ITC<30), class 2 (level 1: $\Delta$st<100 and ITC<100), and class 3 (level 2: $\Delta$st>100 and ITC>100), which represent high-, medium-, and low-quality data, respectively. In addition to the above processing steps, half-hourly flux data were screened using a four-step procedure: (1) data from periods of sensor malfunction were rejected; (2) data collected before or after 1 hr of precipitation were rejected; (3) incomplete 30 min data were rejected when the missing data constituted more than 3% of the 30 min raw record; and (4) data were rejected at night when the friction velocity ($u*$) was less than 0.1 m/s (Blanken *et al*., 1998). There were 48 records per day, with gaps denoted by -6999."

*Snow data*:

*Snow depth* (Sec. 3.3.1, Page 7, Line 258-261):

"In postprocessing, ambient air temperature measured using WXT520 (Vaisala, USA) was used to calibrate the snow depth data (Ryan *et al*. 2008). Data were cross-compared with the measured SWE (introduced in the next subsection), suspicious values were deleted manually followed by noise filtering and, finally, data were averaged to daily output."

*Snow water equivalent* (SWE, Sec. 3.3.2, Page 7, Line 272-276):

"Specifically, SWE data from GMON were calibrated by snow depth and density manually-measured using snow ruler and shovel twice a day (in the mornings and afternoons) in the spring of 2014. To avoid random uncertainties during calibration, a 100 m * 100 m grid around the GMON was designed to measure snow depth at an interval of 10 m (100 measuring spots in the grid). Snow density were also manually-measured within the grid at 6

selected locations.  The averaged snow depth and density were used to fit the coefficients required by the GMON."

*Snow albedo* (Sec. 3.3.3, Page 8, Line 285-287):

"It should be noted that the four-component radiation data (provided in raw format) and the albedo data shown in Figure 4d were calculated by the downward and upward shortwave radiation during 10:00-17:30 (local time) in order to filter out values at high solar zenith angles in early mornings and evenings."

Blowing snow (Sec. 3.3.4, Page 8, Line 296-297):

"To filter the wind noise during the observation (especially in summer), it was necessary to manually delete the suspicious data by comparing the results with the SWE and snow depth data. The data would be rejected when (1) snow depth was zero, (2) wind speed was less than 3 m/s, or (3) air temperature was higher than 10°C."

*Frozen ground data* (Sec. 3.4, Page 8, Line 308):

"The frozen ground data were provided in raw format without any post-processing."

A couple other questions about instruments and data (I focus on snow, as this is my area of expertise): in your figures, only TI rain gages are depicted. I imagine that the Geonor instruments are those located inside the DFIR setups? The TI's are not expected to measure solid precipitation properly, but the Geonors are. However, your data plots show zero or close to zero precip in winter, at the same time when snow depth and water equivalent are positive. Must we conclude that your stations are unable to record solid precipitation?

**Response:** We thank the referee for pointing this out. Actually, unlike the precipitation gauges that were installed inside the DFIR, the Geonor sensors for SWE measurements were located in an open area outside the DFIR to avoid the influence from the fence. In the revised manuscript, the following statements were added (Sec. 3.2.4, Page 6, Line 216-224) to explain more details of the instrument: "In particular, only the precipitation gauge (T200B, Geonor, USA) at the Yakou snow superstation was sheltered with DFIRs to collect both solid and liquid precipitation data. Because the uncertainties of the precipitation gauge (T200B) may result from the unstable voltage or unknown abnormity,

evaporation of the liquid surface, and offset of the instrument, the postprocessing included three steps: (1) manual calibration by adding a certain amount of water into the gauge, (2) abnormal data rejection using the forward-backward filtering (Gustafsson, 1996), and (3) hourly precipitation calculation (using accumulated data before and after each hour). At the other stations, precipitation gauges (TE525M, Texas Electronics, USA) were neither sheltered by Alter shields nor DFIRs. Therefore, only liquid precipitation data were collected. Precipitation data were provided in raw format without any post-processing, which might be underestimated because of the wind and snowfall."

On the other hand, we agree that Figure 4(a) showed zero or close to zero precipitation in winter, at the same time when snow depth and water equivalent are positive (Figure 4b-c), which might correspond to the unique characteristics of the snowfall in the region: 1) in winter (December, January, and February), the snowfall was quite minimum due to the dry westerlies; 2) the snowfall in autumn could accumulate to formulate the snow cover then last in the winter (generally as patchy and shallow snow covers); 3) the snowfall in spring would lead to the maximum snow depth. As we mentioned in the texts, we could measure solid precipitation at the Yakou station. In summary, small snowfall events ultimately led to shallow snowpack observed in the study site, which explained the positive snow depth and SWE observed in Figure 4(b-c).

Then, you talk about snow data, and state that depth and SWE were obtained from the SR50 and the CS725 sensors, respectively. The SPA did not work, apparently, and you link this malfunction to wind and conclude that for this reason snow density is unavailable. However, you do measure depth and SWE with the other sensors! On any case, as this is a dataset paper, I would not expect it to present estimated or derived information (such as density), but only measured data. Additionally, did you make manual depth and SWE measurements with snow probes, samplers or pits? Do these match what was recorded by the sensors?

**Response:** We agree with the referee on this point. The SPA did not work, and we did not install sensors to automatically measure snow density or depth. However, SWE data from GMON were calibrated by snow depth and density manually-measured using snow ruler and shovel twice a day (in the mornings and afternoons) in the spring of 2014. To avoid random uncertainties during calibration,

a 100 m * 100 m grid around the GMON was designed to measure snow depth at an interval of 10 m (100 measuring spots in the grid). Snow density were also manually-measured within the grid at 6 selected locations. The averaged snow depth and density were used to fit the coefficients required by the GMON. The statements were added in Sec. 3.3.2 (Page 7, Line 272-276).

We also provided comparisons between auto-measured (using sensors) and manually-measured data as follows (Figure R1), which, however, were not included in the texts since they were not within the scope of the current manuscript. As shown in Figure R1(a), the trends between the auto-measured SWE (by the GMON) and the manually-measured data were good in general although the GMON outputs seemed to be smoother, which may need further investigation and analysis. On the other hand, the manually-measured snow depth followed with those measured by the SR50 even the SR50 underestimated the snow depth (~4 cm). We think the main reason was the heterogeneity of the snow cover due to the micro-topography and the blowing snow.

[Figure]

(a) SWE

[Figure]

(b) Snow depth

Figure R1. Comparisons between auto-measured (using sensors) and manually-measured data: (a) SWE; (b) Snow depth.

Finally, albedo data looks good, but a bit noisy. Please mention at what solar angle ranges was albedo recorded. Did you filter out values at high angles in the early morning and evening?

**Response:** Good point! The four-component radiation data (provided in raw format) and the albedo data shown in Figure 4d were calculated by the downward and upward shortwave radiation during 10:00-17:30 (local time) in order to filter out values at high solar zenith angles in early mornings and evenings. The explanations were added in Sec. 3.3.3 (Page 8, Line 285-287).

On the other hand, temporary snow covers (several hours to couple of days) during the summer might be another reason causing the noise on the albedo.

**Specific comments:**

L55. Replace "manipulate". Maybe "drive" or "modulate" would be better.

**Response:** We replaced "manipulate" with "influence".

L83: delete "the" before "altitude".

**Response:** We changed to texts to "distributed along an altitudinal gradient".

L85: delete "the" before "alpine".

**Response:** Corrected.